

# Identification of candidate genes associating with soybean cyst nematode in soybean (*Glycine max* L.) using BSA-seq

Haibo Hu[1,2,*], Liuxi Yi[1,*], Depeng Wu[3], Litong Zhang[3], Xuechao Zhou[2], Yang Wu[1], Huimin Shi[1], Yunshan Wei[2] and Jianhua Hou[1]

[1] College of Agriculture, Inner Mongolia Agricultural University, Hohhot, Inner Mongolia, China
[2] Soybean Academy, Chifeng Academy of Agricultural and Animal Husbandry Sciences, Chifeng, Inner Mongolia, China
[3] Collaborative Innovation Center of Regional Modern Agriculture & Environmental Protection Co-constructed by the Province and Ministry, Huaiyin Normal University, Huai'an, China
[*] These authors contributed equally to this work.

Corresponding author
Jianhua Hou, houjh68@163.com, houjh@imau.edu.cn

## ABSTRACT

Soybean cyst nematode disease represents the major soil-borne disease of soybean. Identifying disease-resistant genes in soybean has a substantial impact on breeding of disease-resistant crops and genetic improvement. The present work created the $F_2$ population with the disease-resistant line H-10 and disease-susceptible line Chidou4. 30 respective $F_2$ disease-resistant and disease-susceptible individuals for forming two DNA pools for whole-genome re-sequencing were selected. As a result, a total of 11,522,230 single nucleotide polymorphism (SNPs) markers from these two parental lines and two mixed pools were obtained. Accordng to SNP-index based association analysis, there were altogether 741 genes out of 99% confidence interval, which were mainly enriched into regions of 38,524,128~39,849,988 bp with a total length of 1.33 Mb contain 111 genes on chromosome 2, 27,821,012~29,612,574 bp with a total length of 1.79 Mb contain 92 genes on chromosome 3, 308~348,214 bp with a total of length 0.35 Mb contain 34 genes on chromosome 10, and 53,867,581~58,017, 852 bp with a total length of 4.15 Mb contain 504 genes on chromosome 18. Bulk segregant analysis in $F_2$ generations (BSA-seq) was correlated with a disease resistance interval containing 15 genes. Then, using bioinformatics analysis and differential expression analysis, five candidate genes were identified: *Glyma.02G211400*, *Glyma.18G252800*, *Glyma.18G285800, Glyma.18G287400* and *Glyma.18G298200*. Our results provides a key basis for analyzing the soybean resistance mechanism against soybean cyst nematode and cloning soybean resistance genes.

## INTRODUCTION

Soybean is native to China, which represents a critical economic crop and a feed crop. Soybean cyst nematode disease is a main disease during soybean production. Cyst nematode infects the tender lateral roots of soybean and parasitizes and feeds on the vascular bundles

of soybean roots. Cyst nematodes have a short reproductive cycle and strong reproductive ability, capable of reproducing up to three generations per year. In the case of unsuitable growth environment, cysts in the soil can maintain their viability for 10 years (*Kang et al., 2020*). Soybean cyst nematode disease is widely occurring in soybean producing countries like China, Canada, the United States, Argentina, Brazil, Russia, which can reduce soybean production by about 10% annually worldwide. Soybean cyst nematode disease in China mostly occurs across major soybean production regions in Northeast China and the Yellow River Huai River (*Wang et al., 2020*). Especially in some areas of Northeast China and Inner Mongolia, the Northeast of China has race three of the dominant race, generally leading to a reduction of soybean production by 10% to 30%, with severe cases reaching 70% to 90%, and even crop failure (*Gao et al., 2022*; *Li et al., 2020*). The presence of cyst nematodes in the soil not only makes their eradication challenging but also inflicts significant damage on soybean. While chemical and biological controls can effectively mitigate the damage caused by soybean cyst nematode disease, breeding crops with inherent resistance is an exceptionally efficient strategy for long-term management (*Usovsky et al., 2021*). At present, the genetic background of resistance sources in selected resistant varieties is becoming increasingly narrow. Therefore, carrying out germplasm resource innovation and exploring antagonistic genes is conducive to the efficient utilization of soybean germplasm resources (*Bao et al., 2020*).

Soybean cyst nematode (SCN) resistance has been widely suggested as the quantitative trait regulated *via* diverse genes (*Wang, 2019*). Over 200 quantitative trait loci (QTLs) are mapped onto 20 chromosomes. Typically, *rhg1* and *Rhg4* were subjected to cloning and functional analysis (*Cook et al., 2012*; *Liu et al., 2017*; *Liu et al., 2012*). Many SCN-resistant varieties in the USA are derived from PI88788, which decreases the effective SCN prevention. To breed resistant crops, identifying novel QTL and genes related to disease resistance and broadening genetic foundation to improve soybean resistance to SCN is of great importance. Recently, genes and a complex regulatory network associated with SCN resistance are discovered (*Guo et al., 2020*; *Kofsky, Zhang & Song, 2021*; *Shi et al., 2021*).

Whole genome resequencing of $F_2$ generations through bulk segregant (BSA, *Michelmore, Paran & Kesseli, 1991*) has become a practical target gene mapping method with short experimental cycles and accurate localization. This method has been extensively adopted for gene mapping in various crops, such as sesame (*Miao et al., 2020*), cotton (*Yan et al., 2021*), rice (*Xin et al., 2022*), cucumber (*Pujol et al., 2019*), and wheat (*Zheng et al., 2021*). It has successfully identified the sesame wrinkled leaf trait, sesame plant height and internode length gene *Sidwf1*, cotton fiber lint percentage gene, mature rice plant height (*qPH9*) gene, cucumber virus disease selection genes and wheat epidermal *wax* genes (*Miao et al., 2020*). The combination of whole genome resequencing and BSA can also efficiently and quickly identify target traits such as shade tolerance genes in soybeans (*Zeng et al., 2021b*; *Zhang et al., 2020*), but there are few reports on the application of this method in mining soybean cyst nematode resistant genes. This study is based on the screening results regarding the soybean cyst nematode disease resistant resources in the early stage. Two extreme mixed pools were constructed using SCN resistant variety H-10 and susceptible resource Chidou4 and their $F_2$ populations to sequence the entire genome, locate the

regions significantly associated with soybean cyst nematode disease, and screen candidate genes through molecular bioinformatics methods, aiming to screen genes and loci related to soybean cyst nematode disease. These results provided a basis for molecular breeding regarding to SCN and further elucidating the mechanisms in soybean.

## MATERIALS & METHODS

### Plant materials and phenotype evaluation

A $F_2$ population containing 600 plants was constructed with a cross between Chidou4 (susceptible parental, FI <10%) and H-10 (resistant parental, FI >60%). Chidou4 and H-10 were cultured in Chifeng Academy of Agricultural and Animal Husbandry Sciences, and Shenyang Agricultural University, respectively. For nine differential cultivars, Peking, Pickett, PI90763, PI548316, PI89772, PI209332, PI437654, PI88788, and Lee68, their seeds were provided by Northeast Agricultural University, Key Laboratory of Soybean Biology, Ministry of Education.

H-10, Chidou4, and 600 differential $F_2$ plant were evaluated for their SCN (race 3) resistance within the climate room at Chifeng Academy of Agricultural and Animal Husbandry Sciences. SCN (race 3)-infected soil was added into plastic cups (Ø6 cm × h12 cm), separately. Lee68, Jiyu86, and Liaodou15 were seeded into the SCN-infested soil, cysts were harvested in roots at 30 days later with the 710–250 μm sieve tower from its 250 μm sieve. After washing, cysts were broken with the rubber stopper to collect eggs onto the sieve stack including 250, 75 and 25 μm sieves. Then, we rinsed the mixed sample in 25 μm sieve in the 50 mL plastic conical tube. After adding 50% sucrose solution, eggs were subjected to stirring and 3-min centrifugation at 2,000 rpm, while those in the supernatant or middle layer were harvested onto the 25 μm sieve.

Transplantation of H-10 and Chidou4 was completed, and three replicates, whereas that of 600 differential $F_2$ plants was conducted with no replicate being set. One plant was grown within the plastic cup (Ø6 cm × h12 cm) in one replicate. At five days following transplantation, seedlings were inoculated using 2,000 eggs/cup. Plant growth conditions were 28–24(L:D), 70–80% relative humidity, 16h:8 h (L:D) photoperiod, and daily watering. At 30 days post-inoculation, the nematode cysts in roots of every replicate were harvested to calculate cyst number using the image analysis software (*Wei et al., 2022*).

In this study, female index (FI) was determined and the phenotype data used to analyze QTL, below: FI (%) = (mean cyst number on every line/mean cyst number on Lee68) × 100. Lines were classified into resistant (FI <10), moderately resistant ($10 \leq$ FI <30), moderately susceptible ($30 \leq$ FI <60), or susceptible (FI $\geq$ 60) for classifying SCN response.

### DNA preparation

When conducting disease resistance identification on plants, individual plants were labeled, leaf numbers were taken and stored in liquid nitrogen. Based on the disease resistance results, the corresponding top young leaves of the plants were taken. The genomic DNA of parent materials H-10, Chidou4, and offspring leaves was extracted using the improved cetyltrimethylammonium bromide (CTAB) method (*Zeng et al., 2021b*). The resultant solution was then diluted to the same concentration. Then the DNA of 30 extreme disease

resistant families were mixed with 30 extreme disease susceptible families equally. Two parent pools and two extreme phenotype mixed pools were constructed for whole genome resequencing.

## Library construction and BSA sequencing

Sample library construction and sequencing were completed by Gene Pioneer Biotechnology company ji (Gene Pioneer Biotechnology Co. Ltd., Nanjing, China). The qualified genomic DNA samples were randomly broken into specific sized fragments using Covaris ultrasound, and the fragments were slected using the Agency AMPure XP Medium Kit to concentrate the sample bands around 200–300 bp. The Qubit dsDNA HS AssayKit (500 assays) was used to detect the purified DNA sample size. End repair on the interrupted DNA fragment was performed, and the A base at the 3′ end as well as the specific sequencing primers to the DNA fragment were connected. The connecting products were purified, the appropriate sized DNA fragments were selected, and the DNA fragments on a cBot instrument were amplified to prepare the library. Quality testing was performed on the constructed library, and the qualified library on the machine with an average sequencing depth of $30\times$ for parents and average sequencing depth of offspring $60\times$ were sequenced. The reference genome was Wm82.a2.v1 version of soybean genome (*Song et al., 2016*). SNP detection was achieved using GATK (*McKenna et al., 2010*) tool, and mutation annotation and prediction of mutation impact were conducted with SnpEff software (*Cingolani et al., 2012*).

## Sequencing data analysis

The raw reads obtained from sequencing were filtered, and BWA software was used to compare the filtered sequence numbers with soybean reference genome (https://www.ncbi.nlm.nih.gov/) sequences (*Li & Durbin, 2010*). The results were sorted, the quality was filtered, and duplicates were labeled for subsequent mutation type detection. GATK software was used to search for SNP and Index sites between reference genome and sample. SNP is used for variable filtering parameters in GATK (settings: - filterExpression "QD < 4.0 ‖FS > 60.0‖ MQ < 40.0", - G_filter "GQ < 20", - cluster Windows Size 4). InDel filters through variable filtering parameters (setting: - Filter expression "QD < 4.0 ‖FS > 200.0‖ Read PosRankSum < − 20.0 ‖ Inbreeding coefficient < − 0.8"). ANNOVAR (*Wang, Li & Hakonarson, 2010*) is an efficient software tool for annotating SNPs or InDel based on reference genome GFF3 files. Homozygous SNPs/InDel between two parents should be extracted from the VCF file of SNP/InDel. The reading depth information of homozygous SNPs/InDel above the offspring library can be obtained to calculate the SNP/InDel index.

The calculation of $\Delta$SNP-index was as follows: SNP index (aa) = Maa/(Paa + Maa), SNP index. In the formula, M and P represent parent plants; Maa (Paa) represents the depth of the aa group derived from M (P); And Mab (Pab) refers to the depth of the ab group derived from M (P). We use the genotype of one parent as a reference and use statistical readings from that parent's genotype or other genotypes in the offspring pool. Then, we calculated the ratio of different read counts to the total number, *i.e.,* the SNP/InDel index

of the base site. We filtered out points in both pools where the SNP/InDel index is less than 0.3. Sliding window method was used to present the SNP/InDel index of the entire genome. The average value of all SNP/InDel indexes in each window is the SNP/InDel index of that window. Usually, we use a 1 Mb window size and a 10 kb step size as default settings. The difference in SNP/InDel indices between two pools is calculated as the delta SNP/InDels index. Breakdancer (*McKenna et al., 2010*) software was used to detect SV sites between reference genome and sample. Control FREEC (*Boeva et al., 2012*) software was used to search for CNV regions between reference genome and sample. In order to obtain more accurate mutation information, further filtering was required for each mutation site, and the mutation site annotation software Anovar was used to annotate the location region of the mutation site on the genome.

## Candidate gene localization and analysis

Based on parents and disease-resistant/susceptible single plant mixed pools, the SNP allele frequency of the mixed pool as well as the difference in allele frequency of the same site between two mixed pools were calculated. The positioning interval was obtained. The SNP index of both offspring in the extreme mixing pool, which is less than 0.30 or greater than 0.75 was filter out. An SNP that was not detected in one of the pools was subtracted from the SNP indices of the two extreme mixing pools to obtain a Delta SNP index. Delta SNP index within sliding window was calculated using a sliding window method, and the mean of diverse SNP sites within the sliding window was obtained. The sliding window size was 1.00 Mb in steps of 0.05 Mb. Genes within candidate regions were annotated with the NCBI database (https://www.ncbi.nlm.nih.gov) and clustering analysis was performed using Mercator software (https://www.plabipd.de/mercator_main.html) to classify and calculate the metabolic pathways involved in genes.

## qRT-PCR for candidate genes

The disease-resistant variety H-10 and the susceptible variety Chidou4 was planted in a flowerpot containing SCN Race 3 diseased soil (about 50 cysts/100g soil). After 14 days of emergence, equal samples were taken from the same position at the root of the plant, with a sampling amount of 5 g. This was repeated three times and the samples were quickly placed in liquid nitrogen for freezing. The screened genes were selected, primers were designed, RNA was extracted, and reverse transcription was performed. With *ACTIN7* as the internal reference gene, relative quantitative analysis of the data was performed using the $2^{-\Delta\Delta Ct}$ method. The two-step fluorescence quantitative PCR amplification procedure was conducted as follows: for the amplification curve, 5 min under 95 °C, with one cycle; 10 s under 95 °C, 30 s under 60 °C, 40 cycles. For the melting curve, 15 s under 95 °C, 60 s under 60 °C, and 15 s under 95 °C, and the signal was continuously detected.

# RESULTS

## Phenotypic variation

The plants were cultured as described in the section of materials and methods. There was a significant differences in FI between the parents (Figs. 1A–1B). The average female index

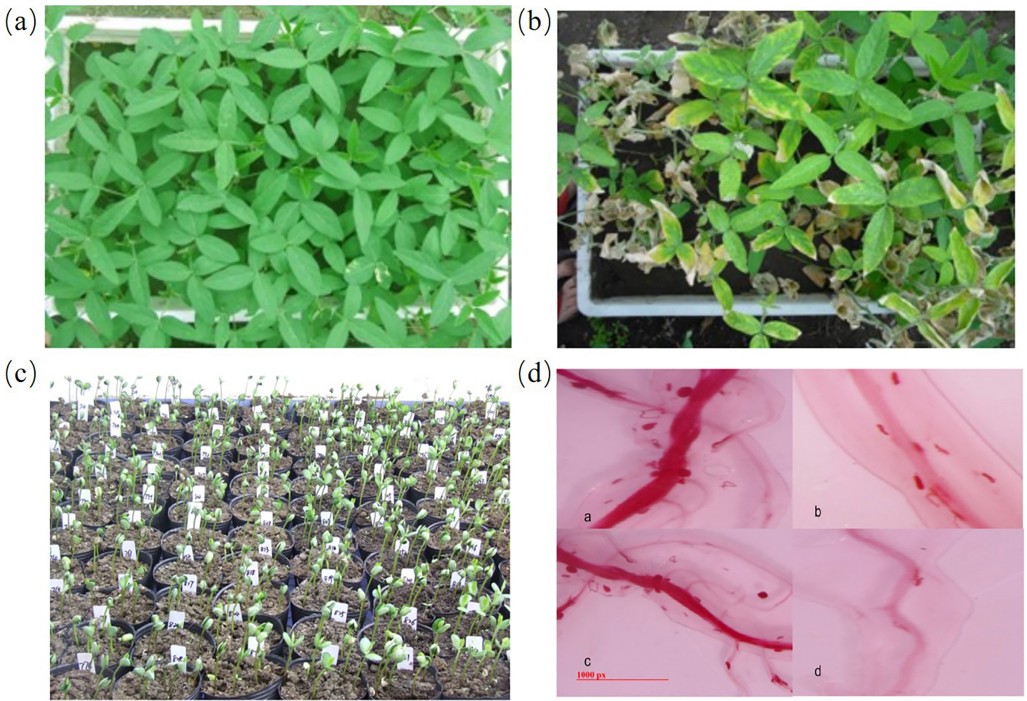

**Figure 1 Morphology of plants and soybean cyst nematode.** At 30 days post-inoculation, the nematode cysts in roots of every replicate were harvested to calculate cyst number using the image analysis software. (A) Phenotyping of disease-resistant parents. (B) Phenotyping of susceptible parents. (C) Phenotyping of $F_2$ population. (D) Morphology soybean cyst nematode using acid fuchsin staining.

of male parent H-10 was 0.61%, making it a highly resistant material; the female index of Chidou4 is 71.28%, indicating high sensitivity. In the $F_2$ population of the hybrid offspring of H-10 and Chidou4 variation range was 0–313.13%, with an average variation rate of 68.02%, between the resistant and susceptible parents (Fig. 1C). The statistical analysis of $F_2$ population resistance level showed a normal distribution. These juvenile nematodes infect the soybean plant's vascular tissue. Female nematodes can be observed at the plant's roots and some of the symptoms of this disease include severe growth retardation, stunting, and a yellowish appearance (Fig. 1D).

## Sequencing quality evaluation

Two DNA hybridization libraries were constructed from the $F_2$ population established from disease resistant parents (H-10) and susceptible parents (Chidou4), as well as 30 disease resistant and susceptible materials. There is a significant difference between the resistance pool (female index <10%) and the susceptibility pool (female index >60%). The entire genome of four samples from a mixed library of parents and offspring was re sequenced using high-throughput sequencing technology, resulting in a total of 112.47 GB of raw sequencing data. After joint filtering, low-quality filtering, and N-filtering using SOAPnuke software, high-quality sequence number data of 110.88Gb was finally obtained. The GC content ranges from 33.95% to 35.32%, indicating high quality sequencing data.

**Table 1  Quality statistics of raw data.**

| Sample | Raw reads | Clean reads | Clean_data rate/% | Clean_GC _Rate/% | Q30/% |
|---|---|---|---|---|---|
| H-10 | 111383826 | 110726851 | 0.99 | 34.66 | 93.46 |
| Chidou4 | 119048756 | 118310801 | 0.99 | 34.59 | 93.59 |
| SCN-sensitive pool | 275551140 | 273558732 | 0.99 | 35.32 | 92.05 |
| SCN-resistance pool | 233209700 | 231785283 | 0.99 | 33.95 | 93.63 |

**Table 2  Matching of quality control data with reference genome.**

| Sample | Raw reads | Mapped reads | Mapping rate/% | GC_ rate/% | Mean depth (X) | Q30/% | Coverage | | |
|---|---|---|---|---|---|---|---|---|---|
| | | | | | | | 1x/% | 5x/% | 10x/% |
| H-10 | 111383826 | 110726851 | 0.99 | 34.66 | 16.97 | 93.46 | 95.58 | 93.02 | 81.43 |
| Chidou4 | 119048756 | 118310801 | 0.99 | 34.59 | 18.14 | 93.59 | 95.51 | 93.24 | 84.15 |
| SCN-sensitive pool | 275551140 | 273558732 | 0.99 | 35.32 | 41.94 | 92.05 | 97.22 | 96.65 | 95.61 |
| SCN-resistance pool | 233209700 | 231785283 | 0.99 | 33.95 | 35.53 | 93.63 | 96.67 | 95.66 | 94.17 |

The average mass value of the vast majority of base sequences is above 30 (Q30 ≥ 92.05%) (Table 1). This suggests an enough data volume for the four samples, reliable sequencing data, qualified sequencing quality, and normal GC distribution, so data can be analyzed in the next step. Following quality control with the soybean reference genome, BWA software (https://bio-bwa.sourceforge.net/) was used to compare filtered sequence numbers from the four samples and Qualimap software was used to compare and statistically analyze the results, The results showed that the genome size was 978,495,272 bp, the effective genome size was 954,832,134 bp (the reference sequence does not contain N). Meanwhile, it was found that the GC content of the reference genome was over 95%, with a comparison rate between 99.28% and 99.41% for the four samples, and with an average sequencing depth of 16.97% (Table 2). The high data comparison rate is conducive to the next step of SNP detection. After sequencing sequence numbers were compared against reference genome, the coverage and sequencing depth distribution of different chromosome regions was calculated on the reference genome. 1× coverage percentage exceeds 95.51%, 5× coverage percentage exceeding 93.02%, 10× coverage percentage exceeds 81.43% (Table 2). By comparing the results with Qualimap software, it was found that the similarity between each sample and the reference gene met the requirements of resequencing analysis. The sequencing data had good coverage depth and coverage, and the reference genome was evenly covered with good randomness, which was conducive to SNP filtering and screening.

## SNP detection and annotation

The analysis results showed that a total of 1,152,230 SNPs were obtained from the four samples, with the least SNPs in the resistant offspring library and the most SNPs in H-10 (resistant parents). SNP loci are distributed on 20 chromosomes of soybean. A total of 128,911 non synonymous SNPs were obtained between the disease resistant material H-10

**Table 3** Annotation of variation site.

| Variation sites information | Chidou4 | H-10 | SCN-sensitive pool | SCN-resistance pool |
|---|---|---|---|---|
| Intronic | 307,215 | 308,529 | 249,065 | 242,056 |
| Intergenic | 2,618,189 | 2,627,581 | 1,904,256 | 1,832,714 |
| Splicing | 704 | 685 | 458 | 489 |
| Upstream | 197,125 | 198,038 | 150,586 | 140,689 |
| Downstream | 165,672 | 165,836 | 125,789 | 896,324 |
| Upstream/Downstream | 13,026 | 12,169 | 8,795 | 9,026 |
| UTR5′ | 32,875 | 32,587 | 19,862 | 19,956 |
| UTR3′ | 29,684 | 28,986 | 24,317 | 23,194 |
| Stop gain | 1,486 | 1,471 | 1,302 | 1,253 |
| Stop loss | 206 | 206 | 186 | 178 |
| Synonymous | 46,852 | 46,917 | 36,819 | 35,892 |
| Non-synonymous | 64,059 | 64,852 | 51,042 | 50,019 |
| SNP numbers | 3,164,285 | 3,186,584 | 248,957 | 248,412 |
| Hom SNP number | 3,267,645 | 3,269,592 | 2,459,562 | 2,269,393 |
| Hete SNP number | 6,624 | 6,663 | 6,303 | 6,237 |
| Hom SNP rate (%) | 99.81 | 99.8 | 99.76 | 99.75 |
| Hete SNP rate (%) | 0.19 | 0.2 | 0.24 | 0.25 |
| TS | 2,125,253 | 2,135,402 | 1,666,723 | 1,556,287 |
| TV | 1,153,424 | 1,147,303 | 906,515 | 835,682 |
| TS/TV | 1.85 | 1.86 | 1.82 | 1.81 |

and the susceptible material Chidou4, and a total of 101,061 non synonymous single nucleotides were obtained from the mixed pool. The statistical data of SNP location and coding information indicate that the number and proportion of SNPs of the same type (between parents and offspring) in the same type of material are usually equal (Table 3). SNP loci are distributed in 15 regions, with the majority located in introns, gene intervals, upstream and downstream of genes, while the rest are distributed in variable splicing sites, synonymous mutations, non synonymous mutations, 3′-untranslated regions, 5′-untranslated regions, and unknown regions. The SNP ratio of homozygous mutations is significantly higher than that of heterozygous mutations, with a SNP ratio of ≥ 99.75% for homozygous mutations.

## Screening and functional analysis of candidate genes

The detected SNPs were analyzed using the SNP index method, the SNP indices of two subsequent libraries were calculated the differences in SNP indices of different chromosome segments in the disease resistant and susceptible libraries were compared, and genetic testing was conducted based on G statistics. When both the $p$-value and the false discovery rate (FDR) are less than 0.001, it was found that the genes controlling disease resistance are mainly concentrated in the 38,524,128–39,849,988 bp fragment (1.33 Mb in length, including 111 genes) on chromosome 2. A total of 741 genes were predicted in the 27,821,012–29,612,574 bp region of chromosome 3 (1.79 Mb in length, containing 92

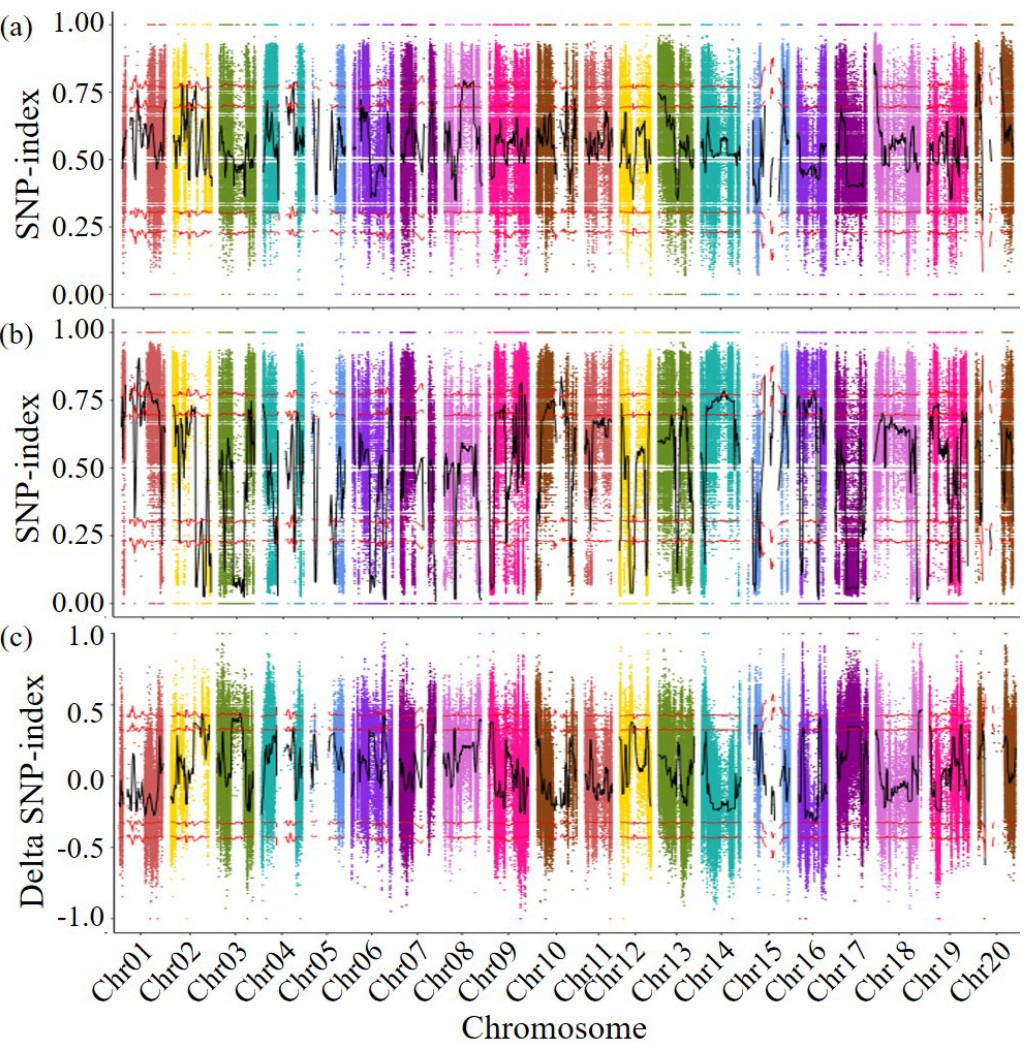

**Figure 2** **Delta SNP-index map of all chromosomes.** The horizontal axis represents the position of chromosomes; The vertical axis represents the Delta SNP index/SNP index value; (A) SNP-index value of resistance pool; (B) SNP-index value of susceptible pool (C) Scatter points in the figure represent the calculated results ΔSNP-index; the black curve represents the corresponding trend line. Above: Correspondingly, the two red lines above are above the 95% and 99% threshold lines, respectively. Below: Correspondingly, the two red lines below are below the threshold of 95% and 99%, respectively.

genes), the 308–348,214 bp region of chromosome 10 (0.35 Mb in length, containing 34 genes), and the 53,867,581–58,017,852 bp region of chromosome 18 (4.15 Mb in length, containing 504 genes) (Fig. 2). A total of 741 genes that may related to the disease resistance traits of soybean cyst nematodes were identified during four specific genomic region. These four genomic regions had been identified as candidate regions for genes related to soybean cyst nematode resistance.

Based on the comparison results and gene functional annotation data in the NCBI database, 351 genes with functional annotations were predicted among four candidate fragments. Through gene functional clustering analysis, it was found that these 351

**Table 4 Gene information in the candidate regions.**

| GeneID | Chromosome | Start | End | Gene Annotation | Homologous gene in Arabidopsis |
|---|---|---|---|---|---|
| Glyma.18G287000 | Chr18 | 56,706,847 | 56,710,004 | Disease resistance protein RPP13. | AT3G46530 |
| Glyma.18G252800 | Chr18 | 53,936,287 | 53,938,433 | Pentatricopeptide repeat-containing protein | AT2G44880 |
| Glyma.18G267900 | Chr18 | 55,199,483 | 55,200,950 | Isoflavone7-O-methyltransferase-like | AT4G35150 |
| Glyma.18G287400 | Chr18 | 56,740,754 | 56,748,259 | 2-oxoglutarate dehydrogenase | AT5G65750 |
| Glyma.18G297200 | Chr18 | 57,444,434 | 57,448,382 | Cytokinin hydroxylase | AT5G38450 |
| Glyma.18G298200 | Chr18 | 57,579,777 | 57,583,443 | 2-alkenal reductase (NADP(+)-dependent) | AT1G65560 |
| Glyma.18G281700 | Chr18 | 56,264,735 | 56,272,703 | Probable disease resistance protein | AT1G09730 |
| Glyma.18G252300 | Chr18 | 53,905,062 | 53,905,983 | Ethylene-responsive element binding protein | AT2G44840 |
| Glyma.18G287100 | Chr18 | 56,710,302 | 56,713,686 | NBS-LRR disease-resistance protein | AT1G53350 |
| Glyma.18G267500 | Chr18 | 55,179,516 | 55,181,798 | Isoflavone 7-O-methyltransferase | AT4G35150 |
| Glyma.18G269500 | Chr18 | 55,320,986 | 55,326,210 | Disease resistance protein (CC-NBS-LRR class) family | AT5G43470 |
| Glyma.18G267800 | Chr18 | 5,192,276 | 55,194,120 | O-methyltransferase family protein | AT4G35150 |
| Glyma.18G281500 | Chr18 | 56,234,929 | 56,241,066 | Disease resistance protein | AT1G09730 |
| Glyma.18G285800 | Chr18 | 56,611,460 | 56,612,978 | Chalcone reductase | AT1G59960 |
| Glyma.02G211400 | Chr02 | 39,675,173 | 39,678,485 | Serine/threonine-protein kinase | AT1G12680 |

genes were annotated into 10 categories (Table 4), including RNA biosynthesis, RNA processing, protein biosynthesis, carbohydrate metabolism, lipid metabolism, enzyme classification, plant hormone action, isoflavone metabolism, chalcone metabolism, and chloroplast redox homeostasis. For instance, Glyma.18G287000, located in Chr18, was found to be associated with disease-resistance RPP13. Glyma.18G252800, also located in Chr18, was found to be associated with pentatricopeptide repeat-containing protein. Glyma.18G267900, also located in Chr18, was found to be associated with isoflavone7-O-methyltransferase-like protein. Glyma.18G287400, also located in Chr18, was found to be associated with 2-oxoglutaratedehydrogenase. Glyma.18G297200 and Glyma.18G298200, both located in Chr18, were found to be associated with Cytokininhydroxylase and 2-alkenalreductase (NADP(+) - dependent), respectively. Glyma.18G281700 was predicted to be associated with probable disease resistance protein. Glyma.18G252300 was predicted to be associated with ethylene-responsive element binding protein. Glyma.18G287100 was predicted to be associated with NBS-LRR disease-resistance protein. Glyma.18G267500 was predicted to be associated with isoflavone7-O-methyltransferase. Glyma.18G269500 was predicted to be associated with disease resistance protein(CC-NBS-LRRclass)family. The last three identified genes in Chr 18 were Glyma.18G267800, Glyma.18G281500 and Glyma.18G285800, which were predicted to be associated with O-methyltransferase family protein, Disease resistance protein and chalconereductase, respectively. The only identified gene located in Chr02 was Glyma.02G211400, which was predicted to be associated with serine/threonine-proteinkinase.

Based on gene functional annotation, the metabolic pathways involved in candidate genes were analyzed, and combined with relevant literature reports, it was found that five genes may be related to soybean disease resistance, namely the protein 2-oxoglutarate

dihydrogenase containing pentapeptide repeat sequences, mitochondria, 2-alkaline reductase (NADP (+) dependent), and chalcone reductase serine/threonine protein kinase. The impact of these candidate genes on soybean disease resistance will be verified in subsequent work. Using vector that can overexpress or silence specific gene to infect plant to verify the effect of gene on soybean disease resistance. However, the resistance might be results from interaction between multiple genes; therefore, more extensive studies need to be designed and conducted conducted to verify our results.

## RT-PCR

Selected the screened genes, design primers, extract RNA, and perform reverse transcription. Using *ACTIN7* as the internal reference gene, perform relative quantitative analysis of the data by $2^{-\Delta\Delta Ct}$ approach. Relative level of maternal parent is shown in Fig. 3, with the maternal parent being 1. The expression levels of selected genes such as pentapeptide repeat protein, 2-hydroxyglutarate dihydrogenase, 2-olefin reductase (NADP (+) dependent), chalcone reductase, and serine/threonine protein kinase increase with the development of soybean cyst nematodes. Based on gene annotation and analysis results, the above five candidate genes exhibit non synonymous mutations in exons at the DNA level, and are highly expressed at the RNA level in roots. Therefore, there is a strong belief that this genes are closely related to the disease resistance of soybean cyst nematodes, especially with the increase of expression levels during soybean cyst nematode infection and development, which can be used for gene cloning, further confirming their biological functions in the development process of soybean cyst nematodes (Fig. 3, Table S1).

## DISCUSSION

### Preliminary gene mapping

The combination of BSA analysis method and high-throughput sequencing technology has shown great advantages in gene localization, and has become the main research strategy for fine mapping of new generation genes. Whole genome re sequencing is carried out on DNA mixed pools constructed by individuals with extreme phenotypes, and fine mapping of target genes is carried out by screening SNP, InDel and other markers closely associated with target traits; the method is extensively used for mining candidate genes for target traits. *Zhang et al. (2020)* used the BSA-Seq method to locate the yellow 622 multi lobule gene in soybean within the 0–250,000 bp, 1,510,000–3,480,000 bp, and 5,570,000–6,710,000 bp segments on chromosome 11, containing a total of 690 genes and identifying 6 potential genes. *Liu et al. (2020)* used BSA-Seq method to discover that the candidate regions for controlling sunflower stem are 0.46 Mb on chromosome 3 and 5 kb on chromosome 14, and screened 8 candidate genes. *Han et al. (2018)* utilized the BSA-Seq method for locating the QTL related to aphid resistance on chromosome 5 of cucumber, containing 43 genes, of which 16 may be related to aphid resistance. *Chen et al. (2021)* used BSA-Seq method for locating two candidate regions for rice blast disease on chromosome 6 at 10,082–11,397 kb and chromosome 11 at 120–266 kb.

*Ochar et al. (2022)* conducted bulk segregant analysis (BSA) and whole-genome resequencing, and obtained 12 potential target trait-related genome regions (20.32 Mb

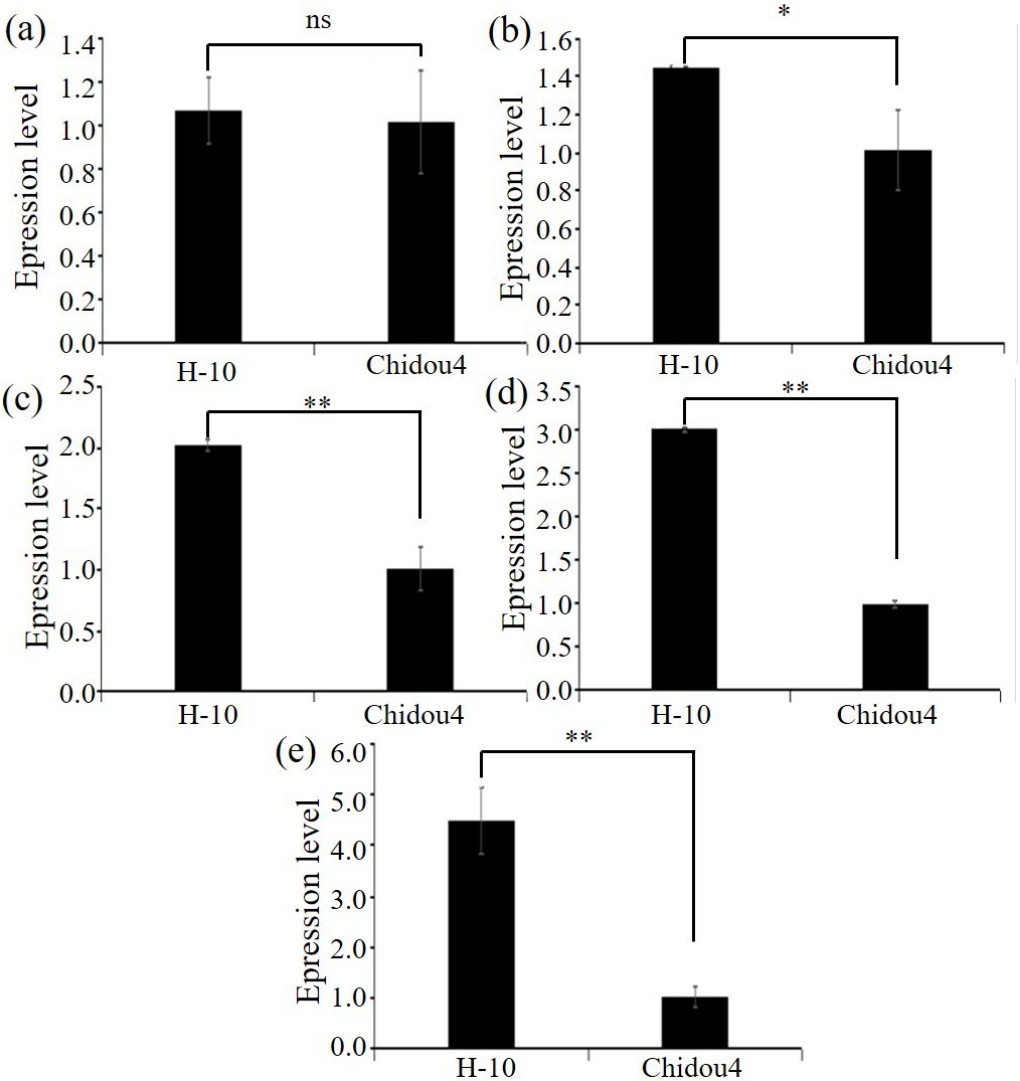

**Figure 3 Expression level of candidate genes between parents.** (A) Expression level of Glyma.18G252800 in contrast varieties. (B) Expression level of Glyma.18G285800 in contrast varieties. (C) Expression level of Glyma.18G287400 in contrast varieties. (D) Expression level of Glyma.18G2298200 in contrast varieties. (E) Expression level of Glyma.02G211400 in contrast varieties. Notes: ** $p < 0.01$. * $p < 0.05$. ns, no significance.

long) by using euclidean distance (ED) correlation algorithm. After comparing sequencing data between mutant and wild-type pools, there was just one single nucleotide mutation (C:G >T:A) in exon 1 of *Glyma.19G207100* linked to soybean crinkled leaf.

*Chen et al. (2022)* used the BSA-Seq method to annotate a total of 143 anthracnose resistant genes on sorghum chromosome 5 within the 1.30 Mb region, including 49 non synonymous mutated genes and 16 frameshift mutated genes. These studies indicate that combining whole genome resequencing with BSA methods can quickly and effectively locate and explore the agronomic traits and stress resistance genes of crops. This study

used H-10 (resistant) and Chidou4 (susceptible) as parents for hybridization to construct an $F_2$ segregation population. Using whole genome resequencing combined with BSA method, the resistance genes controlling the third race of soybean cyst nematode disease were preliminarily located. Research has found that the genes controlling disease-related traits are mainly concentrated in the region of 38,524,128–39,849,988 bp on chromosome 2, 27,821,012–29,612,574 bp on chromosome 3, 308–348,214 bp on chromosome 10, and 53,867,581–58,017,852 bp on chromosome 18, totaling 741 genes. Through gene annotation and metabolic pathway analysis of candidate genes, combined with relevant literature reports, 5 genes have been identified, which are related to soybean disease resistance.

## Function of candidate genes

There are five candidate genes (including pentapeptide repeat protein, 2-hydroxyglutarate dehydrogenase, 2-olefin reductase(NADP(+)dependent), chalcone reductase serine/threonine protein kinase) related to disease resistance within the localization interval of this study.

Serine/threonine protein kinases are an important class of signaling molecules. Upon stressful conditions like salt, pest feeding, drought, or trauma, as well as other cytokines and hormone stimuli, serine/threonine protein kinases quickly phosphorylate and activate at the serine and threonine residues,and further activate downstream signaling molecules through cascade phosphorylation, activating specific signaling pathways, Ultimately, external signals are transmitted to the nucleus, activating or inhibiting specific gene levels (*Zeng et al., 2021a*).

Soybean cyst nematode resistance accounts for the complicated quantitative trait adjusted *via* major and several minor genes. Previous studies on linkage analysis have shown that the main locus of soybean cyst nematode disease is located on chromosomes 08, 11, and 18. For example, the three genes *Glyma18g02580* (amino acid transporter), *Glyma18g02590* (solid NSF attachment protein, SNAP18), and *Glyma18g02610* (wound induced protein WI12) on the highly resistant material Peking are on chromosome 18, whereas *Rhg1* is on chromosome 18 (G-linked group),which controls the resistance of multiple SCN physiological races. *Rgh1* is involved in regulating host plant specific defense responses, Causing obstruction(necrosis) of syncytial development, thereby affecting the development and reproduction of soybean cyst nematodes (*Liu et al., 2012*). The disease resistance gene *Glyma08g11490* (*SHMT*) in PI88788 (resistance) germplasm was located at the *Rhg4* locus and was involved in encoding amino acid transporters $\alpha$-SNAP and WI12 proteins are related to the SCN resistance mechanism (*Liu et al., 2017*). *Li et al. (2016)* found through linkage and association analysis that a QTL SCN3-11 resistant to cyst nematode (race 3) can explain 5.8% of the resistance variation in RIL populations derived from Zhonghuang 13 (susceptible) and Zhongpin 03-5373 (resistant). Aggregating the rhg1 site can significantly improve resistance. Through gene structure and expression analysis, it was further demonstrated that *Glyma.11G234500* at SCN3-11 locus was a new disease resistance gene *GmSNAP11* homologous to *GmSNAP18* at rhg1-b locus (*Lakhssassi et al., 2017*).

According to our findings, resistance site of soybean resistant material H-10 is located within the range of 38,524,128–39,849,988 bp on chromosome 02, while the resistance site on chromosome 18 was enriched within the range of 53,867,581–58,017,852 bp. This may be a new resistance gene enrichment region, and the excavation and functional study of the new genes will provide a basis for the soybean cyst nematode resistance mechanism, and the aggregation of new resistance genes. In the future, overexpressing and silencing vectors could be designed and transformed to soybean to confirm its potential role in plant and nematode interaction.

Researchers have hypothesized that mechanisms contributing to soybean cyst nematode (SCN) resistance are not only species-specific but also genotype-specific (*Kofsky, Zhang & Song, 2021*). Moreover, the culturing environment may also affect the development of SCN resistance. Therefore, under different circumstance such in real fields, different results might be obtained from plants.

## CONCLUSION

Soybean cyst nematode disease is the main soil borne disease of soybean. The identification of soybean disease resistance genes is of great significance for the breeding and genetic improvement of disease resistant crops. In this study, bioinformatics, homologous annotation and differential expression analysis was performed to discover five candidate genes, consisting of pentatricopeptide repeat-containing protein 2-oxoglutarate dehydrogenase, Mitochondrial, 2-alkenal reductase (NADP(+)-dependent), chalcone reductase and serine/threonine-protein kinase. The identification of these candidate genes is a very important advancement in this field, but it also has broad interest in SCN resistance mechanism in soybean research, as this new gene library will help unravel complex mechanisms. In addition, the knowledge gained from these findings will help further subsidize breeding programs, provide information about new candidate genes, and increase knowledge about the genes known to be resistant to the studied nematodes. Ultimately, verifying the true parasitic genes and their subsequent functional characterization from the candidate gene library isolated here will identify weaknesses in the nematode lifecycle, which can serve as targets for new nematode resistance efforts. Obviously, there is still much to understand about how SCN adapts to overcome or evade host plant resistance. Although the community is actively seeking a fully annotated SCN genome. However, RNA sequencing of SCN populations with different reproductive abilities in antagonistic soybeans is promoting comparative transcriptome research to better understand the adaptation aspects of SCN parasitism and antagonistic varieties. The molecular details of nematode virulence will not only help researchers understand plant resistance mechanisms, but also open the door to designing new strategies for resistance and better diagnostic tools to predict population virulence. Evaluating how nematodes adapt to various resistance gene combinations will help determine which resistance gene combinations are most effective and how to best rotate and deploy them.

Using a single source of resistance is not a feasible long-term strategy for managing any pathogen, let alone sexually reproducing nematodes that can easily adapt to resistant varieties. The identification of new disease resistance genes for SCN is the key to overcoming this challenge. The development of molecular markers for molecular breeding based on key genes accelerates the process of breeding disease-resistant varieties and serves as a reference for a complete solution to this problem.

### Funding
This work was supported by the China Agriculture Research System of MOF and MARA (CARS-04-CES09), the Inner Mongolia Autonomous Region Department of Science and Technology and Department of Finance, the Funding for Applied Technology Research and Development (2021GG0144), the Scientific and Technological Projects of Inner Mongolia Autonomous Region (2021GG0374), and the Inner Mongolia Agriculture and Animal Husbandry Youth Innovation Fund (2022QNJJN03). The funders had no role in study design, data collection and analysis, decision to publish, or preparation of the manuscript.

### Grant Disclosures
The following grant information was disclosed by the authors:
China Agriculture Research System of MOF and MARA: CARS-04-CES09.
Inner Mongolia Autonomous Region Department of Science and Technology and Department of Finance.
Funding for Applied Technology Research and Development: 2021GG0144.
Scientific and Technological Projects of Inner Mongolia Autonomous Region: 2021GG0374.
Inner Mongolia Agriculture and Animal Husbandry Youth Innovation Fund: 2022QNJJN03.

### Competing Interests
The authors declare there are no competing interests.

### Author Contributions

- Haibo Hu conceived and designed the experiments, performed the experiments, analyzed the data, prepared figures and/or tables, authored or reviewed drafts of the article, and approved the final draft.
- Liuxi Yi conceived and designed the experiments, performed the experiments, analyzed the data, prepared figures and/or tables, authored or reviewed drafts of the article, and approved the final draft.
- Depeng Wu conceived and designed the experiments, analyzed the data, prepared figures and/or tables, authored or reviewed drafts of the article, and approved the final draft.
- Litong Zhang analyzed the data, prepared figures and/or tables, authored or reviewed drafts of the article, and approved the final draft.

- Xuechao Zhou conceived and designed the experiments, performed the experiments, analyzed the data, prepared figures and/or tables, and approved the final draft.
- Yang Wu conceived and designed the experiments, performed the experiments, analyzed the data, prepared figures and/or tables, and approved the final draft.
- Huimin Shi conceived and designed the experiments, performed the experiments, analyzed the data, prepared figures and/or tables, authored or reviewed drafts of the article, and approved the final draft.
- Yunshan Wei conceived and designed the experiments, performed the experiments, analyzed the data, prepared figures and/or tables, authored or reviewed drafts of the article, and approved the final draft.
- Jianhua Hou conceived and designed the experiments, performed the experiments, analyzed the data, prepared figures and/or tables, authored or reviewed drafts of the article, and approved the final draft.

## Data Availability

The sequences are available at NCBI: PRJNA1046281.

## Supplemental Information

Supplemental information for this article can be found online at http://dx.doi.org/10.7717/peerj.18252#supplemental-information.

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
