# Peer review of "Identification of candidate genes associating with soybean cyst nematode in soybean (Glycine max L.) using BSA-seq"

_PeerJ, doi:10.7717/peerj.18252_

## Round 0.1 · original submission · Major Revisions

Authors are requested to revise the manuscript as per the reviewers' suggestions.

Reviewer 1 ·

Basic reporting

The manuscript writing is clear and straight to the point. There are some errors in the citation format. The data, figures, and tables are acceptable.

Experimental design

It can be accepted and answers all research questions. Technically, it is acceptable and the study is very current. The method is explained in detail but there are grammar errors in the method section. It is advisable to change to simple past tense.

Validity of the findings

The novelty of this study lies in the techniques used and the accuracy of the findings. The data obtained is also reliable. The statistical data is acceptable. The conclusion section should be revised to provide conclusions that address the study objectives directly rather than simply rewriting the results and discussion.

Additional comments

A good manuscript with potential for publication. Please also include relevant pathogen images and the effects on soybean plants infected with the pathogen.

Annotated reviews are not available for download in order to protect the identity of reviewers who chose to remain anonymous.

Reviewer 2 ·

Basic reporting

It is highly recommended to revise the writing professionally, employing past tenses and passive voice to describe methods and results. The experimental procedures must be delineated comprehensively, detailing each step with default or specific parameters, rather than summarized in mere two or three words.

Experimental design

In Figure 2, a considerable proportion of the SNP-index was 1.00 or 0 in both the resistance pool and susceptible pool. These SNP indicate those mutations in the according F2 population pool 100% present the mutations from only one parental sample. The authors should include these mutations in the gene ontology analysis.

In Figure 3, the authors conducted a relative quantitative analysis by comparing it to the reference gene ACTIN7. However, the expression levels of the five selected genes in Chidou4 were consistently indicated as 1, which arose from an erroneous analytical procedure. It is requested to reanalyze the data.

Validity of the findings

An extensive overhaul of the entire manuscript, particularly the Materials Methods, and Results sections, is advised.

The authors should elucidate the differences between ‘upstream and downstream of genes’ and ‘upstream/downstream of genes’

Please include a new section dedicated to the gene functional annotation details.

Regarding the five candidate genes harboring non-synonymous mutations in coding sequences, the author should furnish information on the repercussions of these mutations on the translation process, such as the introduction of single amino acid replacements, premature termination, and frameshifts in open-reading frames.

Reviewer 3 ·

Basic reporting

The manuscript “Identiûcation of candidate genes associating with soybean cyst nematode in soybean (Glycine max L.) using BSA-seq” deals with identifying candidate genes for soybean cyst nematode resistance through whole-genome re-sequencing and association analysis. The use of SNP-index based association analysis, combined with BSA-Seq, offers a robust methodology for pinpointing genomic regions and genes associated with disease resistance.

Comments
• The choice of using the F2 population from the H-10 and Chidou4 lines for whole-genome re-sequencing is appropriate as it allows for the identification of genetic variants associated with disease resistance. However, the size of the F2 population (300 individuals) may be considered small for robust genome-wide association studies, potentially limiting the statistical power of the analysis. Kindly comment on this.
• Could you elaborate on the specific methodologies used for whole-genome re-sequencing and SNP-index based association analysis?
• Integration of genomic selection approaches could enhance the efficiency of breeding for disease resistance by leveraging genomic information across multiple generations. Please discuss on this aspect.
• How were the candidate genes prioritized among the identified pool, and what criteria were used for this prioritization and hat evidence supports the involvement of the identified candidate genes in soybean resistance against soybean cyst nematode?
• What are the potential limitations or confounding factors in the study design that could impact the interpretation of the results? For example, are there any environmental factors or genetic backgrounds of the parental lines that may influence the observed patterns of disease resistance?
• What are the next steps in validating the roles of the identified candidate genes in soybean resistance against soybean cyst nematode?
• Continuous monitoring of soybean cyst nematode populations for the emergence of new virulent strains could inform proactive breeding efforts to maintain durable resistance in soybean cultivars. What are the implications of these findings for sustainable agriculture practices and the long-term management of soybean cyst nematode disease?

Experimental design

Please see section 1

Validity of the findings

Please see section 1

Additional comments

Please see section 1

---

## Round 0.2 · Minor Revisions

The manuscript has been improved and can be accepted after minor revision to address the grammar issues identified by R1.

Reviewer 1 ·

Basic reporting

The author has improved all the suggestions for this section. Please correct the numerous grammar errors throughout this manuscript.

Experimental design

Has been improved by the author

Validity of the findings

The research findings are valid and use the latest techniques.

Additional comments

It can be published after improvements are made, especially in grammar, spelling, and formatting

Annotated reviews are not available for download in order to protect the identity of reviewers who chose to remain anonymous.

Reviewer 2 ·

Basic reporting

NA

Experimental design

NA

Validity of the findings

NA

Additional comments

The advised manuscript is well-prepared for publication. All my concerns have been addressed adequately. I recommend acceptance.

Reviewer 3 ·

Basic reporting

The authors made significant changes in the manuscript.

Experimental design

See section 1

Validity of the findings

See section 1

Additional comments

See section 1

---

## Round 0.3 · Minor Revisions

Dear Authors,

One of the Section Editors had the following final comments which you should respond to. . You are requested to resolve these minor issues and submit a revised, updated version before Acceptance.

"I was not familiar with the term BSA-seq, but after some googling, I found what I think are the original citations.

This citation is needed to define BSA:

Michelmore, R. W., Paran, I., Kesseli, R. V. (1991). Identification of markers linked to disease-resistance genes by bulked segregant analysis: a rapid method to detect markers in specific genomic regions by using segregating populations. Proc. Natl. Acad. Sci. U. S. A. 88, 9828–9832. doi: 10.1073/pnas.88.21.9828

As that seems to be where BSA was first defined.

Combining that with NGS, there are so many that I am not sure who was the original citation.

I do not find the definition of BSA until the discussion, and nowhere do the author's define BSA-Seq. There is opportunity to introduce this term in the abstract and introduction so that the reader doesn't have to do what I had to do to figure it out.

Change Line 31, "Bulk segregant analysis in F2 generations (BSA-seq) was correlated with a disease resistance interval containing 15 genes."

Line 69, "Whole genome resequencing of F2 generations through bulk segregant analysis (BSA, Micklemore et al., 1991) has become a practical target gene mapping method with short experimental cycles and accurate localization." In the remaining paragraph, avoid starting sentences with "It".

More edits are needed for clarification. The title should be "Identification of candidate genes associated with soybean . . . "

Check the remainder of the manuscript for English editing."

---

## Round 0.4 · accepted · Accept

The authors have revised the manuscript as per the suggestions. It can be accepted in its current form.